# Sexually transmitted infections amongst men who have sex with men (MSM) in South Africa

**Rujeko Mashingaidze**[1]*, **Zoe Moodie**[2], **Mary Allen**[3], **Linda-Gail Bekker**[4], **Doug Grove**[5], **Nicole Grunenberg**[2], **Yunda Huang**[2], **Holly E. Janes**[2], **Erica Maxine Lazarus**[1], **Mookho Malahleha**[6], **Maphoshane Nchabeleng**[7], **Fatima Laher**[1]

**1** Faculty of Health Sciences, Perinatal HIV Research Unit, University of the Witwatersrand, Diepkloof, Soweto, South Africa, **2** Vaccine and Infectious Disease Division, Fred Hutchinson Cancer Center, Seattle, Washington, United States of America, **3** Division of AIDS, National Institute of Allergy and Infectious Diseases, National Institutes of Health, Vaccine Research Program, Bethesda, Maryland, United States of America, **4** Desmond Tutu HIV Foundation, University of Cape Town, Cape Town, South Africa, **5** Fred Hutchinson Cancer Center (SCHARP, HVTN), Seattle, WA, United States of America, **6** Synergy Biomed Research Institute, East London, South Africa, **7** Sefako Makgatho Health Sciences University/NHLS, Ga-Rankuwa, South Africa

* rujekomm@gmail.com

**Data Availability Statement:** All data underlying the findings reported in this manuscript are available at: https://atlas.scharp.org/cpas/project/HVTN%20Public%20Data/HVTN%20702/begin.

## Abstract

There is limited data about bacterial STIs in MSM populations in sub-Saharan Africa. Our retrospective analysis used data from the HVTN 702 HIV vaccine clinical trial (October 2016 to July 2021). We evaluated multiple variables. Polymerase chain reaction testing was conducted on urine and rectal samples to detect Neisseria gonorrhoea (NG) and Chlamydia trachomatis (CT) every 6 months. Syphilis serology was conducted at month 0 and thereafter every 12 months. We calculated STI prevalence and the associated 95% confidence intervals until 24 months of follow-up. The trial enrolled 183 participants who identified as male or transgender female, and of homosexual or bisexual orientation. Of these, 173 had STI testing done at month 0, median age was 23 (IQR 20–25) years, with median 20.5 (IQR 17.5–24.8) months follow-up (FU). The clinical trial also enrolled and performed month 0 STI testing on 3389 female participants, median age 23 (IQR 21–27) years, median 24.8 (IQR 18.8–24.8) months FU and 1080 non-MSM males with a median age of 27 (IQR 24–31) years, median 24.8 (IQR 23–24.8) months FU. At month 0, CT prevalence was similar in MSM and females (26.0% vs 23.0%, p = 0.492) but was more prevalent in MSM compared to non-MSM males (26.0% vs 14.3%, p = 0.001). CT was the most prevalent STI among MSM at months 0 and 6 but declined from month 0 to month 6 (26.0% vs 17.1%, p = 0.023). In contrast, NG did not decline in MSM between months 0 and 6 (8.1% vs 7.1%, p = 0.680) nor did syphilis prevalence between months 0 and 12 (5.2% vs 3.8%, p = 0.588). Bacterial STI burden is higher in MSM compared to non-MSM males, and CT is the most prevalent bacterial STI amongst MSM. Preventive STI vaccines, especially against CT, may be helpful to develop.

view?, in a folder named Mashingaidze et al. PLOS Global Public Health 2023.

**Funding:** The authors received no specific funding for this work.

**Competing interests:** The authors have declared that no competing interests exist.

**Abbreviations:** HIV, Human Immunodeficiency Virus; STIs, sexually transmitted infections; MSM, men who have sex with men; TGW, transgender women; CTNG, Chlamydia Trachomatis and Neisseria Gonorrhoea; HICs, high income countries; LMICs, low to middle income countries; PrEP, pre-exposure prophylaxis; PEP, post exposure prophylaxis; RPR, rapid plasmin reagin; TPHA, treponema pallidum hemagglutination assay.

## Introduction

Sexually transmitted infections (STIs) are a global public health concern: 376 million new infections were acquired worldwide in 2016 [1]. In 2017 amongst 15–49 year olds in South Africa, the number of new cases of gonorrhoea was similar in men and women. Although men had more cases than women of Chlamydia trachomatis (CT), (n = 3.9 million vs. n = 1.9 million) and syphilis (n = 47,500 vs n = 23,175) [2], men had fewer new cases of Human Immunodeficiency Virus (HIV) compared to women (n = 92,000 vs. n = 107,000) [3].

Data from Europe, the USA and China show that people who are born male and have sex with other people born male, regardless of their gender identity or sexual orientation (MSM), have a high burden of HIV and other STIs [4]. For example, in England in 2019, there were 77,371 new STI diagnoses in MSM [5]. Of the new cases recorded amongst MSM in 2019, NG (n = 33,853; 44%) and CT (n = 23,187; 30%) were the most common. In the same year there were 5,875 (8%) new cases of syphilis in MSM [5]. In comparison, there were 97,450 new STI diagnoses amongst non-MSM males [5]. Chlamydia was more common amongst non-MSM males (n = 46,192; 47.4%) followed by NG (n = 15,253; 15.7%). There were 1,016 (1%) new syphilis infections.

The number of new gonorrhea cases diagnosed in 2019 (n = 70.936) was the largest annual number reported since records began in 1918 and 33 853 (47.7%) of those cases were reported in MSM [5].

There is limited research about STIs in MSM in South Africa. Most research is centred around HIV. There is poor health-seeking behavior amongst MSM (accessing HIV/STI screening and testing services, and disclosure of sexual practices to healthcare workers) within South Africa and other Sub-Saharan countries [6]. The fear of stigma and discrimination by healthcare workers poses as a barrier to access HIV/STI services [6,7]. The criminalization of homosexuality in some African countries also poses as a barrier to seeking healthcare services [7]. Political, social and religious factors play an important role in limiting research in this key population. Estimating the burden of STIs in MSM is therefore a huge challenge.

In our analysis, we describe the STI prevalence amongst MSM enrolled in a preventive HIV vaccine trial which had routinely tested for sexually transmitted infections as part of study procedures.

## Methods

### Ethics statement

Ethics approval and consent to participate: Ethical approval was obtained from the University of the Witwatersrand Human Research Ethics Committee (ethics reference number: 160208B). All participants provided written informed consent.

We retrospectively analysed data collected from 18 October 2016 to 21 July 2021 in the HVTN 702 HIV vaccine efficacy clinical trial.

### Setting

The HVTN 702 phase 2b/3 trial of ALVAC-HIV and gp120 protein boost candidate HIV preventive vaccine was conducted at 14 trial sites in South Africa [8]. The trial enrolled from October 2016 to June 2019 with the study protocol specifying enrolment of 30%-35% men. After an interim data safety and monitoring board meeting in January 2020, non-efficacy was declared and vaccinations were stopped [9].

### Eligibility for HVTN 702 and trial procedures

To be enrolled into HVTN 702, participants had been assessed as being healthy, 18-35-year-old HIV-uninfected adults at risk of HIV acquisition who were willing to receive HIV

counselling and testing (S1 Protocol). The final decision to either include or exclude a volunteer from the study was based on laboratory test results, medical history, physical examination findings and answers to questionnaires. Possible barriers to retention e.g. employment were also considered.

During the trial, polymerase chain reaction (PCR) for Chlamydia trachomatis (CT) and Neisseria gonorrhoea (NG) was conducted on urine samples for all participants, in addition rectal swabs were collected for MSM and vaginal swabs were collected for all female participants every six months. Syphilis serology using rapid plasma reagin (RPR) with reflex treponema pallidum hemagglutination assay (TPHA) was performed every year. Routine STI testing had been introduced in protocol version 2, on 8 December 2016 and the first STI testing data was collected on 8 May 2017, therefore STI data were unavailable for 10 MSM trial participants at month 0.

The protocol had provisioned that sites offered participants the best available standard of prevention throughout the trial, including risk reduction counselling at all visits, provision of male and female condoms, information regarding male circumcision and referral for voluntary medical male circumcision (VMMC), and referrals for pre- and post-exposure prophylaxis (PrEP and PEP) according to local guidelines.

## Data collection

From the HVTN 702 database, we selected the data of enrolled participants born male who identified at the baseline visit as either male or transgender female and sexual orientation as homosexual or bisexual on the demographic questionnaire. For this analysis, we defined these individuals as MSM.

We collated demographic, clinical, behavioural and microbiological data. Demographic variables, which were collected at baseline, included date of birth, race, sex at birth, gender identity, sexual orientation, marital status and level of education. Clinical data variables collected at baseline included male circumcision status and pre-existing STIs. Behavioural data variables collected at baseline were number of sexual partners, frequency of condom use, transactional sex, and alcohol intake associated with sex. We analysed CT and NG results from months 0, 6, 12, 18 and 24, and syphilis results from months 0, 12 and 24. Indeterminant STI results were considered as missing data. We did not analyse data beyond month 24, because it coincided with significant reduction in data collection especially after study product vaccinations were stopped for non-efficacy.

## Statistical analysis

Descriptive statistics were used to summarize the demographic, clinical, and sexual behaviour data. STI prevalence was estimated as the empirical fraction of participants with a positive test result with two-sided 95% confidence intervals (CI) calculated using the Wilson score method [10]. Baseline STI prevalence were compared between gender subgroups by Barnard's exact test. As risk reduction was anticipated to be greatest by the first post-baseline assessment [11–13], STI prevalence was compared at month 0 and month 6 (month 12 for syphilis) using generalized estimating equations with logit link and independent correlation structure to model the binary outcome of presence of an STI (yes/no). Descriptive rather than inferential statistics were reported for STI prevalence at the later time points due to the higher rates of missing data. Differences in demographics and reported risk behaviours at month 0 among MSM with at least one STI at months 0 and/or 6 vs. those without an STI at months 0 and 6 were assessed by Barnard's exact test (2 category variables) or Fisher's exact test (3 or more category

variables). R statistical software (version 4.0.4, R Foundation for Statistical Computing, Vienna, Austria) was used for statistical analysis.

## Results

Of the 9918 participants screened in HVTN 702, 2745 had been assigned male sex at birth and of those, 1608 enrolled in the study. At the baseline visit, 1598 reported their gender identity as male and 10 as transgender female, 183/1608 (11.4%) reported their sexual orientation as homosexual or bisexual.

With regards to MSM participants, the population had a median age of 22 (IQR 20–25), most identified as males (n = 173/183; 94.5%), as Black race (n = 179/183; 97.8%), resided in an urban/city/town (n = 167/183; 91.3%) and had completed high school as the highest level of education (n = 99/183; 54.1%) (Table 1). Participants were enrolled from 5 of the 9 provinces in South Africa, namely Gauteng (n = 131/183; 71.6%), Western Cape (n = 24/183; 13.1%), KwaZulu-Natal (n = 13/183; 7.1%), North West (n = 12/183; 6.6%) and Eastern Cape (n = 3/183; 1.6%).

Data from baseline physical examinations showed that 90/183 (49.2%) of MSM participants had been circumcised.

At month 0, 134/183 (73.2%) MSM participants reported to be married or with a main sex partner, and 20/183 (10.9%) reported living with their main sex partner. 39/183 (21.3%) reported that their main sex partner had other sex partners. MSM participants recalled that in the preceding month, they had had a median number of 8 (IQR 4–19) sexual acts with a median number of 2 (IQR 2–6) sexual partners. Inconsistent condom use was reported by 149/183 (81.4%), 122/183 (66.7%) had engaged in unprotected sex following alcohol intake, and 73/183 (39.9%) had engaged in transactional sex.

CT was the most prevalent STI among MSM at months 0 and 6 (Fig 1 and Table 2). Compared to month 0, CT prevalence declined at month 6 (26.0% vs 17.1%, p = 0.023). CT prevalence was 12.8%, 14.0%, 17.1% at months 12, 18, and 24 respectively (Fig 1).

Compared to month 0, NG prevalence was similar at month 6 (8.1% vs 7.1%, p = 0.680) with prevalence of 6.4%, 5.1%, 8.5% at the later time points, month 12, 18 and 24 respectively (Fig 1). Syphilis prevalence was similar at months 0 and 12 (5.2% vs 3.8%, p = 0.588).

STI reinfections were also present in the MSM cohort. Overall, there were 41 unique participants that accounted for the 50 participants/STI combinations that had multiple positives across the time-points months 0, 6, 12, 18, 24.

At month 0, CT prevalence was similar in MSM and females (26.0% vs 23.0%, p = 0.492) but was more prevalent in MSM compared to non-MSM males (26.0% vs 14.3%, p = 0.001). A similar pattern existed for NG (Table 3). At month 0, syphilis prevalence was higher in MSM compared to females (5.2% vs 1.3%, p = 0.019) and also higher than in non-MSM males (5.2% vs 1.6%, p = 0.007). A similar pattern existed for co-infection of CT and NG. At month 0, co-infection of CT and NG was more prevalent in MSM than females (7.5% vs 2.5%, p = 0.019), and significantly more prevalent in MSM compared to non-MSM males (7.5% vs 1.2%, p = <0.001).

Between months 0 and 6, 70/173 (40.5%) of MSM had STIs (Table 4). Among MSM with STIs, multiple sexual partners (>2) were common in 41/70 (58.6%), transactional sex was observed in 30/70 (42.9%) and unprotected sex with alcohol was observed in 44/70 (62.9%) irrespective of alcohol quantity. There was evidence that MSMs aged ≤ 24 years were more likely to have at least one STI between months 0 and 6 compared to older MSMs (unadjusted p = 0.038). There is not convincing evidence of a difference by other variables.

**Table 1. Demographic and sexual characteristics for all MSM participants and MSM, non-MSM male and Female participants who had STI data available at Month 0.**

| | Cohort (N) | | | |
|---|---|---|---|---|
| | **All MSM (183)** | **MSM with Mo 0 STI data (173)** | **Non-MSM Mo 0 STI data (1080)** | **Females Mo 0 STI data (3389)** |
| **Demographic characteristic** | N (%) | N (%) | N (%) | N (%) |
| *Age in years* Median (IQR) | 22 (20–25) | 23 (20–25) | 27 (24–31) | 23 (21–27) |
| *Race* | | | | |
| Black | 179 (97.8) | 169 (97.7) | 1062 (98.3) | 3359 (99.1) |
| Coloured/Mixed | 4 (2.2) | 4 (2.3) | 13 (1.2) | 20 (0.6) |
| Multiple races | 0 (0.0) | 0 (0.0) | 3 (0.3) | 3 (0.1) |
| White | 0 (0.0) | 0 (0.0) | 1 (0.1) | 5 (0.1) |
| Indian | 0 (0.0) | 0 (0.0) | 0 (0.0) | 2 (0.1) |
| Asian | 0 (0.) | 0 (0.0) | 1 (0.1) | 0 (0.0) |
| *Gender Identity* | | | | |
| Male | 173 (94.5) | 164 (94.8) | 1070 (99.2) | 2 (0.1) |
| Transgender female | 10 (5.5) | 9 (5.2) | 0 (0.0) | 0 (0.0) |
| Female | 0 (0.0) | 0 (0.0) | 6 (0.6) | 3386 (99.0) |
| Transgender male | 0 (0.0) | 0 (0.0) | 0 (0.0) | 1 (0.0) |
| Gender variant | 0 (0.0) | 0 (0.0) | 2 (0.2) | 0 (0.0) |
| Prefer not to answer | 0 (0.0) | 0 (0.0) | 2 (0.2) | 0 (0.0) |
| *Sexual Orientation* | | | | |
| Homosexual | 123 (67.2) | 116 (67.1) | 9 (0.8) | 27 (0.7) |
| Bisexual | 60 (32.8) | 57 (32.9) | 1 (0.1) | 56 (1.7) |
| Heterosexual | 0 (0.0) | 0 (0.0) | 1067 (98.8) | 3307 (97.6) |
| Not sure | 0 (0.0) | 0 (0.0) | 2 (0.2) | 2 (0.1) |
| Prefer not to answer | 0 (0.0) | 0 (0.0) | (0.1) | 0 (0.00) |
| *Marital Status* | | | | |
| Married/Main partner | 134 (73.2) | 128 (74.0) | 939 (86.9) | 3303 (88.6) |
| No main partner | 46 (25.1) | 42 (24.3) | 112 (10.4) | 220 (6.5) |
| Unknown | 3 (1.6) | 3 (1.7) | 29 (2.7) | 166 (4.9) |
| *Lives with Spouse/Main Partner* | | | | |
| Yes | 20 (10.9) | 18 (10.4) | 207 (19.2) | 453 (13.4) |
| No | 114 (62.3) | 110 (63.6) | 731 (67.7) | 2550 (75.2) |
| Unknown | 3 (1.6) | 3 (1.7) | 30 (2.8) | 166 (4.9) |
| N/A – [No main partner] | 46 (25.1) | 42 (24.3) | 112 (10.4) | 220 (6.5) |
| *Highest education completed* | | | | |
| Some primary school | 0 (0.0) | 0 (0.0) | 5 (0.5) | 11 (0.3) |
| Completed primary school | 0 (0.0) | 0 (0.0) | 27 (2.5) | 38 (1.1) |
| Some high school | 36 (19.7) | 34 (19.7) | 505 (46.8) | 1510 (44.6) |
| Completed high school | 99 (54.1) | 94 (54.3) | 447 (41.4) | 1556 (45.9) |
| National certificate | 3 (1.6) | 3 (1.7) | 5 (0.5) | 28 (0.8) |
| Some tertiary school | 39 (21.3) | 37 (21.4) | 81 (7.5) | 225 (6.6) |
| Completed tertiary | 6 (3.3) | 5 (2.9) | 9 (0.8) | 16 (0.5) |
| Some graduate school | 0 (0.0) | 0 (0.0) | 0 (0.0) | 4 (0.1) |
| Completed graduate school | 0 (0.0) | 0 (0.0) | 1 (0.1) | 0 (0.0) |
| Prefer not to answer | 0 (0.0) | 0 (0.) | 0 (0.0) | 1 (0.0) |
| *Residence* | | | | |
| Urban/City/Town | 167 (91.3) | 157 (90.8) | 882 (81.7) | 2707 (79.9) |

*(Continued)*

**Table 1.** (Continued)

| | Cohort (N) | | | |
|---|---|---|---|---|
| | **All MSM (183)** | **MSM with Mo 0 STI data (173)** | **Non-MSM Mo 0 STI data (1080)** | **Females Mo 0 STI data (3389)** |
| Rural/Countryside | 13 (7.1) | 13 (7.5) | 170 (15.7) | 521 (15.4) |
| Unknown | 3 (1.6) | 3 (1.7) | 28 (2.6) | 161 (4.8) |
| *Number of sexual partners in last 30 days* | | | | |
| **Median (IQR)** | 2 (2–6) | 3 (2–6) | 2 (1–3) | 2 (1–2) |
| *Number of sexual acts in last 30 days* | | | | |
| **Median (IQR)** | 8 (4–19) | 8 (4–19) | 8 (5– 15) | 7 (4–12) |
| *Transactional sex in last 30 days* | | | | |
| Yes | 73 (39.9) | 72 (41.6) | 158 (14.6) | 725 (21.4) |
| No | 109 (59.6) | 100 (57.8) | 918 (85.0) | 2653 (78.3) |
| Unknown | 1 (0.5) | 1 (0.6) | 4 (0.40) | 11 (0.3) |
| *Unprotected sex with alcohol use in last 30 days* | | | | |
| 1–2 | 52 (28.4) | 50 (28.9) | 330 (30.6) | 868 (25.6) |
| 3–5 | 47 (25.7) | 43 (24.9) | 219 (20.3) | 482 (14.2) |
| ≥6 | 23 (12.6) | 22 (12.7) | 138 (12.8) | 189 (5.6) |
| Never | 61 (33.3) | 58 (33.5) | 390 (36.1) | 1848 (54.5) |
| Unknown | 0 (0.0) | 0 (0.0) | 0 (0.0) | 2 (0.1) |

## Discussion

Our study results show that, in South Africa, CT is the most prevalent STI amongst MSM. Furthermore, CT and NG are each more prevalent in MSM compared to non-MSM males, but approximate the burdens in females. Syphilis prevalence was significantly higher in MSM compared to both females and non-MSM males.

At month 0, CT prevalence was more than triple that of NG amongst MSM. These results are consistent with STI data from the Sibanye study which piloted HIV prevention interventions for MSM in South Africa from 2015 to 2016 [14]. However, other regions have shown different trends. For example, in England, NG was the most prevalent STI amongst MSM [5]. According to a study conducted in five cities amongst MSM in the USA (2017), the prevalence of STIs was dependent on the anatomical location, with rectal CT being more common than rectal NG (7.3% vs 4.5%, p = <0.001) [15].The high CT prevalence globally can be attributed to the predominant asymptomatic presentation of the infection permitting unrecognized persistent infection and onward transmission [16].

During each visit of the HVTN 702 trial, all participants received an HIV and STI care package which represents an ideal rather than the actual standard of care for the general population in South Africa. The prevention package included behavioral risk reduction counseling, advocacy and referral for medical male circumcision as appropriate, free condoms and lubricant, regular STI testing and treatment referral or access, as well as oral PEP/PrEP counseling, referral and access. By way of comparison, in the South African public healthcare sector, patients would generally not be scheduled to receive counselling multiple times per year, and would not receive routine or even symptom-driven diagnostic STI tests. We observed in this study that CT prevalence amongst MSM declined significantly in the first 6 months, which could be interpreted as a measure of initial success of this care package. However, CT prevalence did not continue to decline thereafter. And despite the care package, the prevalence of NG and syphilis were unchanged to month 24. Taken together, these data suggest that even in

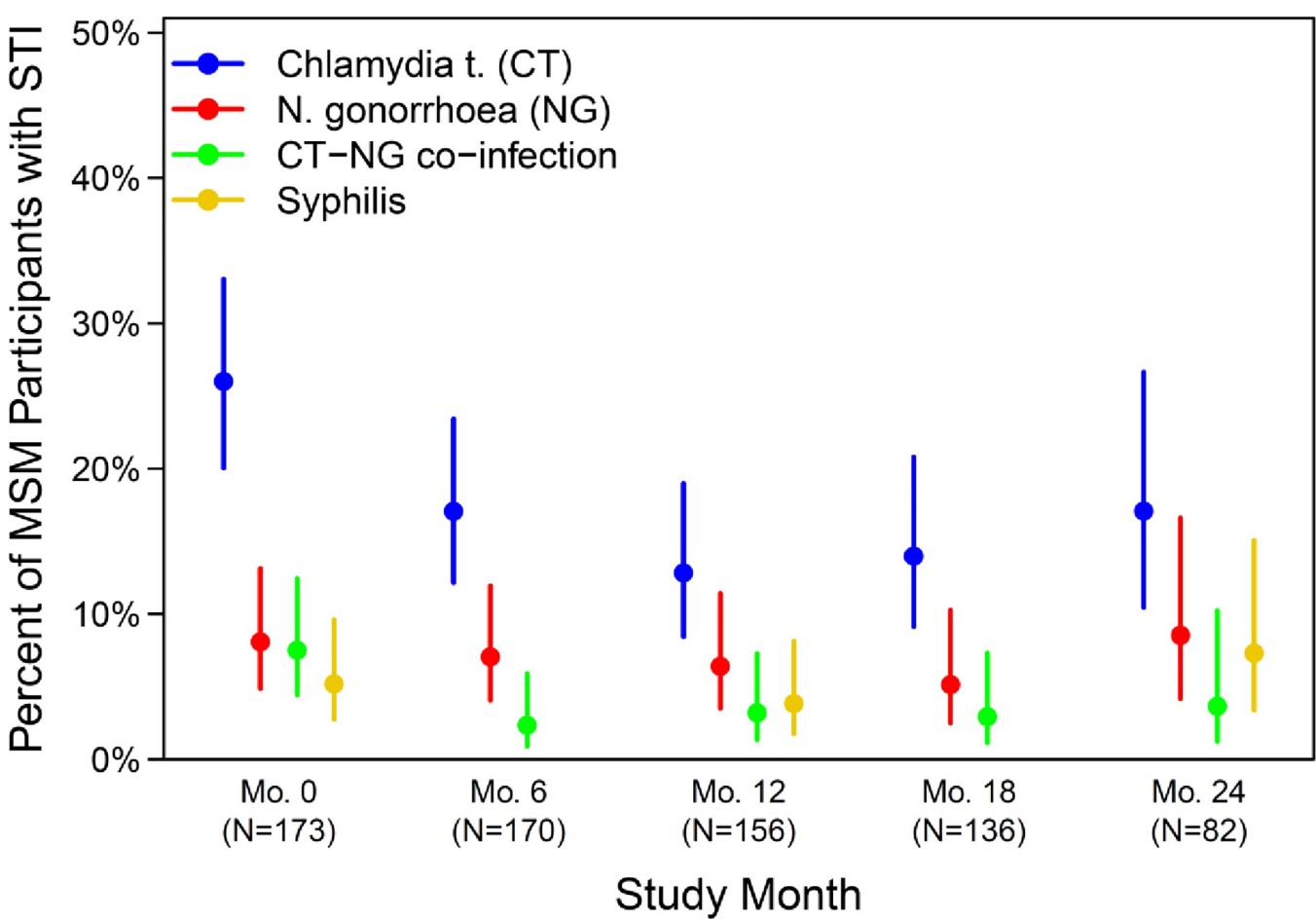

**Fig 1. Estimated STI prevalence and 95% Cis over time among MSMs with STI testing data available in HVTN 702.**

the ideal context of behavioral and medical care within a clinical trial, STI prevalence amongst MSM remains similar over time, notwithstanding an initial decline in CT prevalence.

A moderate number of participants had CT and NG co-infection at month 0. Risky sexual behavior (e.g. multiple sexual partners) has been proven to predispose MSM to multiple

**Table 2. STI prevalence over time of enrolled MSM participants with available STI testing data.**

| STI prevalence (n, %, CI95) | Month 0 | Month 6 | Month 12 | Month 18 | Month 24 |
|---|---|---|---|---|---|
| Number of participants who completed the visit (n) | 183 | 171 | 156 | 136 | 82 |
| Number of participants tested (n) | 173 | 170 | 156 | 136 | 82 |
| Chlamydia trachomatis | 45 (26.0) (20.0–33.0) | 29 (17.1) (12.1–23.4) | 20 (12.8) (8.5–19.0) | 19 (14.0) (9.1–20.8) | 14 (17.1) (10.5–26.6) |
| Neisseria gonorrhoea | 14 (8.1) (4.9–13.1) | 12 (7.1) (4.1–11.9) | 10 (6.4) (3.5–11.4) | 7 (5.1) (2.5–10.2) | 7 (8.5) (4.2–16.6) |
| Chlamydia/Neisseria co-infection | 13 (7.5) (4.4–12.4) | 4 (2.4) (0.9–5.9) | 5 (3.2) (1.4–7.3) | 4 (2.9) (1.1–7.3) | 3 (3.7) (1.3–10.2) |
| Syphilis | 9 (5.2) (2.8–9.6) | N/A | 6 (3.8) (1.8–8.1) | N/A | 6 (7.3) (3.4–15.1) |

**Table 3. Baseline (month 0) STI testing results for females at birth, non-MSM males at birth and MSM who had data available.**

| | Female at birth (n = 3389) | Male at birth | | p-value[3] for MSM vs. Female at birth | p-value[3] for MSM vs. non-MSM |
|---|---|---|---|---|---|
| | | non-MSM (n = 1080) | MSM (n = 173) | | |
| **C trachomatis[1]** | | | | 0.492 | **0.001** |
| Positive | 779 (23.0) | 154 (14.3) | 45 (26.0) | | |
| Negative | 2541 (75.0) | 900 (83.3) | 125 (72.3) | | |
| Not done/indeterminate | 69 (2.0) | 26 (2.4) | 3 (1.7) | | |
| **N gonorrhea[1]** | | | | 0.120 | **< 0.001** |
| Positive | 179 (5.3) | 25 (2.3) | 14 (8.1) | | |
| Negative | 3134 (92.5) | 1034 (95.7) | 156 (90.2) | | |
| Not done/indeterminate | 76 (2.2) | 21 (1.9) | 3 (1.7) | | |
| **C trachomatis/ N gonorrhea[1] co-infection** | | | | **0.019** | **< 0.001** |
| Positive | 86 (2.5) | 13 (1.2) | 13 (7.5) | | |
| Negative | 3239 (95.6) | 1047 (96.9) | 157 (90.8) | | |
| Not done/indeterminate | 64 (1.9) | 20 (1.9) | 3 (1.7) | | |
| **Syphilis[2]** | | | | **0.019** | **0.007** |
| Positive | 44 (1.3) | 17 (1.6) | 9 (5.2) | | |
| Negative | 3322 (98.0) | 1056 (97.8) | 161 (93.1) | | |
| Not done/indeterminate | 23 (0.7) | 7 (0.6) | 3 (1.7) | | |

[1]Test performed on cervical/vaginal swab, urine, or rectal swab.

[2]Both non-treponemal and treponemal test must be positive for a positive diagnosis.

[3]Barnard's exact test—based on Positive and Negative result categories only.

concurrent STIs [15,17]. In our study, participants reported a median number of 2 sexual partners, which was stable between baseline and subsequent time points. However, the stated number of sexual partners may be subject to under-reporting owing to recall and social desirability bias.

Our analysis found that STI acquisition factors such as multiple sexual partners, transactional sex and unprotected sex associated with alcohol intake, were not statistically linked to STI prevalence (p value > 0.05). However these factors have been previously linked with increased prevalent STI risk in African men [18]. In the USA, MSM who reported multiple sexual partners were found to have an increased risk of incident bacteria STIs [19]. Individuals who transact sex may be unable to negotiate safe sex for fear of losing out on transactional benefits e.g. money or groceries [20]. Although alcohol intake has been linked with risky sexual behavior [21], other authors have found little evidence that alcohol intoxication during sex affected condom use, especially in individuals who used condoms when sober [22].

The syndromic approach to STI management, which is the current standard of care in South Africa and other low and middle income countries, relies on patients presenting with signs and symptoms for presumptive diagnosis to treat STIs, without the use of laboratory tests [23]. The CDC, however, recommends screening MSM every 3–6 months for bacterial STIs [24].

There are some limitations of this analysis. We did not include the anatomical site (e.g. urogenital or rectal) or presence/absence of symptoms in our analysis. However, it has already been well established that most STIs are asymptomatic [1,15,23]. For our analysis, CT and NG results among MSM were each considered positive if either the urine or rectal sample testing was positive. Including the specific anatomical site for the positive test was therefore not a

**Table 4.** STI acquisition factor analysis of demographic and month 0 behavioural risk variables for MSM sub-population with at least one STI at months 0 and/or 6 vs. those without an STI at months 0 and 6.

| | MSM without an STI at months 0 and 6 (103) | MSM with at least one STI at months 0 and/or 6 (70) | [1]p value |
|---|---|---|---|
| **Demographic characteristic** | N (%) | N (%) | |
| *Age in years* | | | 0.038 |
| ≤ 24 | 64 (62.1) | 54 (77.1) | |
| > 24 | 39 (37.9) | 16 (22.9) | |
| *Race* | | | 0.15 |
| Black | 99 (96.1) | 70 (100.0) | |
| Coloured/Mixed | 4 (3.9) | 0 (0.0) | |
| *Gender Identity* | | | 1.00 |
| Male | 98 (95.1) | 66 (94.3) | |
| Transgender female | 5 (4.9) | 4 (5.7) | |
| *Sexual Orientation* | | | 1.00 |
| Homosexual | 69 (67.0) | 47 (67.1) | |
| Bisexual | 34 (33.0) | 23 (32.9) | |
| *Marital Status* | | | 0.26 |
| Married/Main partner | 80 (77.7) | 48 (68.6) | |
| No main partner | 22 (21.4) | 20 (28.6) | |
| Unknown | 1 (1.0) | 2 (2.9) | |
| *Lives with Spouse/Main Partner* | | | 0.42 |
| Yes | 10 (9.7) | 8 (11.4) | |
| No | 70 (68.0) | 40 (57.1) | |
| Unknown | 1 (1.0) | 2 (2.9) | |
| N/A – [No main partner] | 22 (21.4) | 20 (28.6) | |
| *Highest education completed* | | | 0.27 |
| Some high school | 16 (15.5) | 18 (25.7) | |
| Completed high school | 61 (59.2) | 33 (47.1) | |
| National certificate | 1 (1.0) | 2 (2.9) | |
| Some tertiary school | 21 (20.4) | 16 (22.9) | |
| Completed tertiary | 4 (3.9) | 1 (1.4) | |
| *Residence* | | | 0.07 |
| Urban/City/Town | 98 (95.1) | 59 (84.3) | |
| Rural/Countryside | 4 (3.9) | 9 (12.9) | |
| Unknown | 1 (1.0) | 2 (2.9) | |
| *Number of sexual partners in last 30 days* | | | 1.00 |
| ≤ 2 | 43 (41.7) | 29 (41.4) | |
| > 2 | 60 (58.3) | 41 (58.6) | |
| *Number of sexual acts in last 30 days* | | | 0.61 |
| ≤ 7 | 47 (45.6) | 28 (40.0) | |
| > 7 | 56 (54.4) | 42 (60.0) | |
| *Transactional sex in last 30 days* | | | 0.74 |
| Yes | 42 (40.8) | 30 (42.9) | |
| No | 61 (59.2) | 39 (55.7) | |
| Unknown | 0 (0.0) | 1 (1.4) | |
| *Unprotected sex with alcohol use in last 30 days* | | | 0.84 |

*(Continued)*

**Table 4.** (Continued)

| | MSM without an STI at months 0 and 6 (103) | MSM with at least one STI at months 0 and/or 6 (70) | [1]p value |
|---|---|---|---|
| **1-2** | 32 (31.1) | 18 (25.7) | |
| **3-5** | 26 (25.2) | 17 (24.3) | |
| **$\geq$ 6** | 13 (12.6) | 9 (12.9) | |
| **Never** | 32 (31.1) | 26 (37.1) | |
| *Anal Sex in last 30 days* | | | 0.21 |
| **Yes** | 91 (88.3) | 66 (94.3) | |
| **No** | 12 (11.7) | 4 (5.7) | |

[1]Barnard's Exact Test was performed on variables that had 2 categories, while Fisher's Exact Test was performed on variables with 3 or more categories.

requirement for our analysis. Urine testing for non-MSMs was regarded as sufficient for detection of STIs in this population. Rectal testing was excluded in non-MSMs, as the risk of receptive anal sex with a male partner was considered nil based on their behavioral questionnaires. CT and NG testing amongst females was done on both urine and vaginal swabs.

Due to differences in sexual practices amongst MSM, females and non-MSMs, including the specific anatomical site for the positive test was therefore not a requirement for our analysis.

However, the difference in diagnostic tests in each group is a potential for differential diagnostic bias. This is likely to affect those non-MSMs who self-identified as heterosexual and failed to disclose engaging in receptive anal sex with a male partner for fear of being labeled gay or MSM. Failure to do rectal testing in these participants would result in missed/undiagnosed STIs.

Knowledge of the anatomical site of STIs is vital in clinical practice, as this will guide healthcare workers on which systems to examine routinely during consultations and which specimen to collect for laboratory testing. In a systematic review on extragenital infections, it was shown that among MSM, the prevalence of rectal NG and CT ranges from 0.2% to 24% and 2.1% to 23%, respectively, and prevalence of pharyngeal NG and CT ranges from 0.5% to 16.5% and 0% to 3.6%, respectively [25]. Another limitation of our analysis is that oropharyngeal STI testing was not conducted during the HVTN 702 trial. Pharyngeal infections with NG or CT are a potential source of urethral infections [25,26]. Approximately 70% of gonococcal and chlamydial infections might be missed if urogenital-only testing is performed among MSM [26]. Pharyngeal testing by nucleic acid amplification test (NAAT) for NG and CT is recognized as an important sexual health consideration for MSM [26].

Following non-efficacy declaration, there was a reduction in study follow up visits. This was the largest contributor to the reduction in number of participants and STI data over time. HIV infection, study dropouts and missed visits also contributed to the loss of study participants and data. There is potential for differential loss to follow up (LTFU) bias as participants who either tested positive for HIV or an STI dropped out of the study. This would result in an underestimation of the prevalence of STIs at later time points. A strength of this study is the longitudinal data, permitting the identification of patterns over time to inform the sexual health needs of MSM.

## Conclusions

Our study supports other findings that CT is the most prevalent STI amongst MSM, and indeed also in non-MSM males and females, in South Africa. Furthermore, we found that

bacterial STI burden is higher in MSM compared to non-MSM males, which suggests the need for healthcare workers to tailor care for MSM. The type of tailored care for MSM, however, may need to exceed what is the current ideal standard of care, because in our study STI prevalence amongst MSM remained similar over time, although there was an initial decline in CT prevalence. When one considers these data against the backdrop of the global trends of increasing antibiotic resistance and the discovery of fewer novel antibiotics, preventive STI vaccines, especially against CT, would be helpful to develop.

## Supporting information

**S1 Protocol. HVTN 702 eligibility criteria.** Inclusion and exclusion for HVTN 702 clinical trial (page 92–96). Available from: https://clinicaltrials.gov/ProvidedDocs/49/NCT02968849/Prot_ICF_000.pdf.
(PDF)

## Author Contributions

**Investigation:** Mary Allen, Linda-Gail Bekker, Doug Grove, Yunda Huang, Holly E. Janes, Erica Maxine Lazarus, Mookho Malahleha, Maphoshane Nchabeleng, Fatima Laher.

**Methodology:** Mary Allen, Holly E. Janes, Erica Maxine Lazarus, Maphoshane Nchabeleng, Fatima Laher.

**Supervision:** Fatima Laher.

**Writing – original draft:** Mary Allen, Doug Grove, Nicole Grunenberg, Yunda Huang, Holly E. Janes, Erica Maxine Lazarus, Mookho Malahleha, Fatima Laher.

**Writing – review & editing:** Rujeko Mashingaidze, Zoe Moodie, Mary Allen, Linda-Gail Bekker, Doug Grove, Nicole Grunenberg, Yunda Huang, Holly E. Janes, Erica Maxine Lazarus, Mookho Malahleha, Maphoshane Nchabeleng, Fatima Laher.

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
