## [Decision Letter · Decision Letter 0]

26 Jul 2022

PGPH-D-22-00629

Sexually transmitted infections amongst men who have sex with men (MSM) in South Africa

Dear Dr. Mashingaidze,

Thank you for submitting your manuscript to PLOS Global Public Health. After careful consideration, we feel that it has merit but does not fully meet PLOS Global Public Health’s publication criteria as it currently stands. Therefore, we invite you to submit a revised version of the manuscript that addresses the points raised during the review process.

EDITOR:

Dear authors,

Reviewers and I consider that your article is relevant. Kindly see the comments by Reviewer 1 and 2 which highlight issues with the manuscript. We invite you to submit a revised version of the manuscript that addresses the points raised during the review process.

We look forward to receiving your revised manuscript.

Kind regards,

Zulma Vanessa Rueda, M.D. Ph.D.

Academic Editor

Journal Requirements:

2. In the online submission form, you indicated that "Availability of data and materials: The data that support the findings of this study are available on request from the corresponding author, RM. Data is currently not available to the public due to publication restrictions". All PLOS journals now require all data underlying the findings described in their manuscript to be freely available to other researchers, either 1. In a public repository, 2. Within the manuscript itself, or 3. Uploaded as supplementary information.

3. Please provide separate figure files in .tif or .eps format and removed from the manuscript file.

Additional Editor Comments (if provided):

Dear Authors,

Kindly see the comments by Reviewer 1 and 2 which highlight issues with the manuscript. We invite you to submit a revised version of the manuscript that addresses the points raised during the review process.

Reviewers' comments:

Reviewer's Responses to Questions

**Comments to the Author**

1. Does this manuscript meet PLOS Global Public Health’s publication criteria? Is the manuscript technically sound, and do the data support the conclusions? The manuscript must describe methodologically and ethically rigorous research with conclusions that are appropriately drawn based on the data presented.

Reviewer #1: Yes

Reviewer #2: Yes

2. Has the statistical analysis been performed appropriately and rigorously?

Reviewer #1: No

Reviewer #2: Yes

3. Have the authors made all data underlying the findings in their manuscript fully available (please refer to the Data Availability Statement at the start of the manuscript PDF file)?

Reviewer #1: Yes

Reviewer #2: No

4. Is the manuscript presented in an intelligible fashion and written in standard English?

Reviewer #1: Yes

Reviewer #2: Yes

5. Review Comments to the Author

Reviewer #1: • STI testing appears to have been done on rectal and urine samples (for men), but only one set of results is reported? Were these always concordant? This is listed as a limitation in the analysis, but surely it must be possible to compare in men which samples were positive? This must have been recorded?

• Very few details are given re: how the MSM were recruited. Did 702 use a recruiting procedure targeted at MSM, or did the overall strategy just happen to include some MSM? Obviously the MSM are much smaller, so the gender/sex comparisons, while certainly of interest, are limited in terms of MSM sample size and unknown generalizability.

• Since a major part of the paper is comparing participants by gender & sex, it is an odd choice not to include the demographic, behavioural, and clinical data in Table 1. There is no way to know how these populations are comparable apart from sex & gender.

• The hypothesis for why M0 and M6 would be different and the reason for a GEE between those 2 time points is unclear. Furthermore, why compare by sex & gender at M0 when STIs were presumably run on all 3 groups all throughout the trial? A GEE that compared the 3 groups at all of the time points would likely make more sense. This study cohort also provides an opportunity to measure incidence, but this does not appear to have been undertaken & it’s unclear why not.

• Why was a risk factor analysis not carried out for STI prevalence within MSM? Even an unadjusted table could help to profile who is at most risk within the MSM sub-group. The discussion contains a lot of mention of prior risk factor analyses, but it’s unclear why these weren’t repeated in this study?

• With respect to the STI rates not declining at later time points, could this be due to re-infection (assuming results are being transmitted to participants and treatment is occurring)?

Reviewer #2: Thank you for the opportunity to review this interesting manuscript. Using results from a discontinued trial on HIV vaccine efficacy among healthy, HIV negative adults aged 18-35 years from 14 sites in South Africa, the authors reported the prevalence of chlamydia, gonorrhea, and syphilis amongst a sub-population of men who have sex with men (MSM) who participated in trial. The authors were able to measure prevalence at baseline, and at 6, 12, 18, and 24 months for chlamydia and gonorrhea. After an initial decrease in chlamydia infections from baseline to 6 months, prevalence remained stable for the duration of the study. Gonorrhea infections mostly stayed the same across time periods. While this manuscript offers some interesting data on an issue where there are substantial gaps in the literature, I feel there are a number of shortcomings that should be addressed by the authors. Suggestions are below, but the two most important ones, in my opinion, are how the authors dealt with loss to follow-up (by Month 24, over 50% of their original sample did not have samples available); and describing the inclusion/exclusion criteria, and diagnostic protocols for women and non-MSM in their study. The authors invest substantial time in describing differences in prevalence between MSM, women and non-MSM men; however, very little attention is paid to describing the comparison group(s) – in particular, which women and non-MSM were swabbed for chlamydia, gonorrhea, and syphilis, and how this may have impacted findings.

Suggestions and Comments

Abstract: Results – says 183 participants who identified as male or transgender female & of homosexual or bisexual orientation. But in Background, the authors state that their goal was to compare rates to females and non-MSM males. Should clarify that they compared results of the 183 MSM to the rest of the participants in a larger trial.

Introduction

• Lines 41-42: can the authors comment on the prevalence of STIs in non-MSM populations – this comparison can help bolster their argument regarding the excess burden of STIs among MSM.

• Line 45: - would suggest quantifying what “majority” means.

• Lines 47-48: While I don’t disagree with this assessment, the authors should back this assertion with some references. Additionally, there has not been a complete lack of research in sub-Saharan Africa (SSA), so I’d suggest a more thorough literature review on presentation of STI rates in the SSA context. As well, this would be a good place to specify some of the reasons behind the lack of STI research in the SSA context. For example, the high costs of diagnostics, laboratory capacity, barriers to access and accessibility – including specific stigma against MSM in sub-Saharan countries.

Methods

• Line 58: Would suggest “The trial enrolled participants from October 2016…with the study protocol specifying…”

• The authors do not report any information regarding data collection and analysis among females and non-MSM males. In fact, from Line 78 onwards, a reader would be lead to believe that only data on MSM were analysed. This needs to be clarified, as a comparison to females and non-MSM participants from the larger trial comprises a critical component of the manuscript. Others have found higher prevalence of STI in rectal samples from MSM (Jones et al, 2020. JIAS Vol. 23, S6) in South Africa; thus without knowledge of how samples are obtained from women and (to a certain extent non-MSM), this may not be exactly an apples-to-apples comparison.

• It looks like the authors ignored indeterminate results (Table 3) – this needs to be specified in the Methods section, especially given the lower absolute number of infections, and their large lost to follow-up.

• Lines 95-96: With respect to the generalized estimating equations model – was this a logistic regression? And what is the rationale for using GEE as opposed to a simpler method, such as McNemar’s test especially if the authors are only comparing 2 points in time. Relatedly, did the authors consider using GEE across all points in time where data were available? If not, why not?

As this was an analysis of data from an HIV vaccine trial, I assume that ethics and informed consent covered publication of results from secondary outcomes?

Results

• Lines 126-132: Again, a reminder to the authors that a clarification regarding comparisons to females and non-MSM males is strongly suggested. As well, what happens to differences upon adjustment by age, region, etc.?

• Table 2: 95%CI would be useful here.

• Additionally, by month 24 less than half of their original study (or only 82 of the original 173) were not available for analysis. What were the reasons for availability/unavailability? This becomes crucial when trying to interpret results, as prevalence is impacted by dropouts. More information is needed from authors on how they dealt with this issue.

Discussion

• Subanalyses: did the authors examine who cleared CT infections, and who did not? It is interesting that CT declined from Month 0 to Month 6, but GC did not. This might suggest some natural clearing of CT or perhaps false positives in Month 0.

• Lines 184-186: Why did the authors not examine whether these factors were associated with STI prevalence?

• Limitations – were syphilis tests able to discern acute infections from historical ones?

6. PLOS authors have the option to publish the peer review history of their article (what does this mean?). If published, this will include your full peer review and any attached files.

**Do you want your identity to be public for this peer review?** For information about this choice, including consent withdrawal, please see our Privacy Policy.

Reviewer #1: No

Reviewer #2: No

---

## [Decision Letter · Decision Letter 1]

24 Nov 2022

PGPH-D-22-00629R1

Sexually transmitted infections amongst men who have sex with men (MSM) in South Africa

Dear Dr. Mashingaidze,

Thank you for submitting your manuscript to PLOS Global Public Health. After careful consideration, we feel that it has merit but does not fully meet PLOS Global Public Health’s publication criteria as it currently stands. Therefore, we invite you to submit a revised version of the manuscript that addresses the points raised during the review process.

Your manuscript has been assessed by a new reviewer, who has provided their own feedback on the work, as well as assessing whether the comments from the previous round were adequately addressed. As you will see from the comments, the reviewer acknowledges that the manuscript has improved, but there remain some important concerns which should be addressed before it is suitable for publication.

We look forward to receiving your revised manuscript.

Kind regards,

Joseph Donlan

Editorial Office

Journal Requirements:

Additional Editor Comments (if provided):

Reviewers' comments:

Reviewer's Responses to Questions

**Comments to the Author**

1. If the authors have adequately addressed your comments raised in a previous round of review and you feel that this manuscript is now acceptable for publication, you may indicate that here to bypass the “Comments to the Author” section, enter your conflict of interest statement in the “Confidential to Editor” section, and submit your "Accept" recommendation.

Reviewer #3: (No Response)

2. Does this manuscript meet PLOS Global Public Health’s publication criteria? Is the manuscript technically sound, and do the data support the conclusions? The manuscript must describe methodologically and ethically rigorous research with conclusions that are appropriately drawn based on the data presented.

Reviewer #3: Partly

3. Has the statistical analysis been performed appropriately and rigorously?

Reviewer #3: No

4. Have the authors made all data underlying the findings in their manuscript fully available (please refer to the Data Availability Statement at the start of the manuscript PDF file)?

Reviewer #3: Yes

5. Is the manuscript presented in an intelligible fashion and written in standard English?

Reviewer #3: Yes

6. Review Comments to the Author

Reviewer #3: This revised manuscript describes changes over time in STI prevalence during follow-up in a clinical trial aimed at evaluating efficacy of an HIV vaccine in South Africa. Data of this kind are not common in sub-Saharan Africa. The authors responded adequately to the reviewers’ comments; however, there are some outstanding issues worth addressing.

The authors only provide a statistical comparison of STI prevalence based on months 0 and 6 (justified by the fact that very few data were missing at these time points). Yet, they report STI prevalence up until month 24 and infer on changes in STI prevalence during the 24-month period. There needs to be consistency here. Please report STI prevalence and test for changes in this prevalence for the same follow-up period. If the authors are so concerned with violating the MCAR assumption of the GEE model, why not use a different model that relaxes this assumption (e.g., mixed effect model with random intercept)? (As an aside, I do agree with the authors that modeling STI incidence would be somewhat difficult without a follow-up time prior to baseline.)

The authors seem to have data on behaviors during follow-up (ln 123-129). If so, it is unclear why they did not also evaluate the changes in behavior over time to corroborate their findings of changes in STI prevalence. These need to be added to the analysis, if available.

The analysis on risk factors for STI is ambiguous (Table 5) and the footnotes do not adequately describe what was being modeled as an outcome. Any STI within 24 months? And risk factors evaluated at which time points (as the authors state “between months 0-24”, ln 148)? If the covariates were time-updated, they could simply model the outcome STI at each time point along with corresponding covariates measured at the same time points. Techniques to account for repeated measured (i.e., GEE or mixed-effects) could be applied accordingly. Regardless, this analysis needs to be described in more detail in the statistical analysis section.

The reviewers asked for more clarity in the statistical analysis section and the authors did not accommodate these requests accordingly (even though their answerers were sufficient). Please state the endpoint being modeled. This is a logistic regression model with general estimating equations, please state as such (if this model was indeed entertained).

One final general comment: please adhere to the STROBE reporting guidelines. There are several items that appear to be missing in this manuscript.

Minor comments:

- ln 14. Please provide totals for all groups along with their median (IQR) follow-up time. (Potential for differential LTFU should also be assessed in the limitations section.)

- ln 52. What type of “health-seeking behavior” is being referenced here? Also, there are no citations to support this claim.

- ln 58. In what ways are the “profiles” being described? “changes in STI prevalence” would be more appropriate.

- Table 1. Why include a column on all MSM? Since the main outcome is STI prevalence, would it be more obvious to exclude those without an STI measurement? Or did those without STI measured at month 0 have an STI measured at a later time point? The inclusion and exclusion criteria could be more thoroughly described.

- Table 3 is excessive. The authors could very well add a sentence, such as: “XX CT infectious were observed in NN individuals.” to begin the description of prevalence over time.

- ln 252-256. The authors seem to be missing the point of the reviewer. Normally, if you test in more places, you are more likely to find an STI. Is there a differential diagnostic bias between MSM, females at birth and non-MSM? For example, if the non-MSM never engaged in insertive anal sex with a male partner, their risk of having an anal STI is essentially nil, hence no reason to test for rectal STIs. The potential for differential diagnostic bias would likely depend on whether MSM was correctly classified in this instance. Please modify the discussion accordingly.

7. PLOS authors have the option to publish the peer review history of their article (what does this mean?). If published, this will include your full peer review and any attached files.

**Do you want your identity to be public for this peer review?** For information about this choice, including consent withdrawal, please see our Privacy Policy.

Reviewer #3: No

---

## [Decision Letter · Decision Letter 2]

13 Mar 2023

Sexually transmitted infections amongst men who have sex with men (MSM) in South Africa

PGPH-D-22-00629R2

Dear Dr Mashingaidze,

We are pleased to inform you that your manuscript 'Sexually transmitted infections amongst men who have sex with men (MSM) in South Africa' has been provisionally accepted for publication in PLOS Global Public Health.

Best regards,

Siyan Yi, MD, MHSc, PhD

Academic Editor

Reviewer Comments (if any, and for reference):

Reviewer's Responses to Questions

**Comments to the Author**

1. If the authors have adequately addressed your comments raised in a previous round of review and you feel that this manuscript is now acceptable for publication, you may indicate that here to bypass the “Comments to the Author” section, enter your conflict of interest statement in the “Confidential to Editor” section, and submit your "Accept" recommendation.

Reviewer #3: All comments have been addressed

2. Does this manuscript meet PLOS Global Public Health’s publication criteria? Is the manuscript technically sound, and do the data support the conclusions? The manuscript must describe methodologically and ethically rigorous research with conclusions that are appropriately drawn based on the data presented.

Reviewer #3: Yes

3. Has the statistical analysis been performed appropriately and rigorously?

Reviewer #3: Yes

4. Have the authors made all data underlying the findings in their manuscript fully available (please refer to the Data Availability Statement at the start of the manuscript PDF file)?

Reviewer #3: Yes

5. Is the manuscript presented in an intelligible fashion and written in standard English?

Reviewer #3: Yes

6. Review Comments to the Author

Reviewer #3: Thank you for the very clear answers to my queries.

7. PLOS authors have the option to publish the peer review history of their article (what does this mean?). If published, this will include your full peer review and any attached files.

**Do you want your identity to be public for this peer review?** For information about this choice, including consent withdrawal, please see our Privacy Policy.

Reviewer #3: No
